# Thermoplastic Vulcanizates with an Integration of High Wear-Resistant and Anti-Slip Properties Based on Styrene Ethylene Propylene Styrene Block Copolymer/Styrene Ethylene Butylene Styrene Block Copolymer/Solution-Polymerization Styrene-Butadiene Rubber

**DOI:** 10.3390/polym16152221

**Published:** 2024-08-04

**Authors:** Zhicheng Li, Jianbin Xiao

**Affiliations:** Key Laboratory of Rubber-Plastics (Ministry of Education), School of Polymer Science and Engineering, Qingdao University of Science and Technology, Qingdao 266042, China; li353033831@163.com

**Keywords:** solution-polymerized styrene butadiene rubber (SSBR), thermoplastic elastomer (TPE), wear resistant, anti-slip properties, dynamic vulcanization

## Abstract

Distinguished from traditional vulcanized rubber, which is not reusable, thermoplastic elastomer (TPV) is a material that possesses both the excellent resilience of traditional vulcanized rubber and the recyclability of thermoplastic, and TPVs have been widely studied in both academia and industry because of their outstanding green properties. In this study, new thermoplastic elastomers based on solution polymerized styrene butadiene rubber (SSBR) and thermoplastic elastomers (SEPSs/SEBSs) were prepared by the first dynamic vulcanization process. The high slip resistance and abrasion resistance of SSBR are utilized to improve the poor slip resistance of SEPSs/SEBSs, which provides a direction for the recycling of shoe sole materials. In this paper, the effects of different ratios of the rubber/plastic phase (R/P) on the mechanical properties, rheological properties, micro-morphology, wear resistance, and anti-slip properties of SSBR/TPE TPVs are investigated. The results show that the SSBR/TPE TPVs have good mechanical properties. The tensile strength, tear strength, hardness, and resilience of the TPVs decrease slightly with an increasing R/P ratio. Still, TPVs have a tensile strength of 18.1 MPa when the ratio of R/P is 40/100, and this reaches the performance of the vulcanized rubber sole materials commonly used in the market. In addition, combined with microscopic morphology analysis (SEM), it was found that, with the increase in the R/P ratio, the size of the rubber particles gradually increased, forming a stronger crosslinking network, but the rheological properties of TPVs gradually decreased; crosslinking network enhancement led to the increase in the size of the rubber particles, and the increase in the size of rubber particles made the material in the abrasion of rubber particles fall easily, thus increasing its abrasion volume. Through dynamic mechanical analysis and anti-slip tests, when the R/P ratio was 40/100, the tan δ of TPVs at 0 °C was 0.35, which represents an ordinary vulcanized rubber sole material in the market. The viscoelasticity of TPVs increased with the increase in the R/P ratio, which improved the anti-slip performance of TPVs. SSBR/TPE TPVs are expected to be used in footwear and automotive fields due to their excellent abrasion resistance and anti-slip performance.

## 1. Introduction

A special type of high-performance material in high-performance thermoplastic elastomers (TPEs) is called thermoplastic vulcanizate elastomers (TPVs), which consist of crosslinked rubber as a dispersed phase homogeneously dispersed in a continuous-phase thermoplastic elastomer by dynamic vulcanization (DV) [1]. The rubber particles are dispersed in a continuous resin matrix under a strong shear. The continuous plastic is like the ocean, and the rubber particles are like the small islands in the ocean. This results in the microstructure of the “sea-island” phase. Due to their unique structure, thermoplastic vulcanizates offer not only the elasticity of traditional vulcanized rubber, but they also provide the recyclability and melt processability of thermoplastics [2,3,4]. Thus, thermoplastic vulcanizates (TPVs), which are typically recyclable materials replacing non-recyclable thermoset rubbers, have attracted a lot of attention because of their environmental friendliness and fast growth; they are being widely used in the automotive industry, construction materials, shoe sole materials. etc. [5,6,7,8]. As an example, most of the shoe sole materials are made of traditional thermosetting vulcanized rubber, and the waste rubber materials have a great impact on the ecological environment. Therefore, it is urgent to develop a recyclable material that can replace traditional vulcanized rubber.

The microstructure of thermoplastic vulcanizates greatly influences the mechanical, elastic, and rheological properties of thermoplastic elastomers, which are among the most important properties of elastomers [9]. The content of the rubber phase in the TPV, the degree of crosslinking, the size of the rubber phase, the uniformity of dispersion of the rubber particles, and the structure of the rubber crosslinking network all play a key role in the mechanical properties and elasticity of TPVs [10,11,12,13]. In general, the smaller the size of the rubber phase, the more uniformly the rubber is dispersed in the thermoplastic elastomer and the better the overall performance of the prepared thermoplastic elastomer. Therefore, the selection of more compatible plastic and rubber phases leads to a better preparation of thermoplastic elastomers, since a more homogeneous two-phase structure can be formed.

In 1981, Monsanto Company (ExxonMobil AES) developed the first commercially available thermoplastic vulcanizate elastomers based on blends of ethylene propylene diene rubber (EPDM) and polypropylene (PP) [14,15]. Until now, EPDM/Polypropylene TPVs remain the best-known TPV products and are commercially used in a wide range of applications, including the automotive, construction, and electronics industries [16,17,18].

Unfortunately, traditional TPVs such as EPDM/PP TPVs cannot meet some special requirements, such as the need for both good wear resistance and high skid resistance in shoe sole materials [19]. Styrene-based thermoplastic elastomers (SBSs, SISs, SEBSs, and SEPSs) have good wear resistance and excellent comprehensive mechanical properties [20]. However, their skid resistance is poor and cannot meet the skid resistance requirements of shoe sole materials. SSBR combines the rolling resistance and wet skid resistance of rubber and is a copolymer of butadiene and styrene that is similar in structure to styrene-based thermoplastic elastomers [21,22].

The microstructure and properties of thermoplastic vulcanizates prepared by dynamic vulcanization are affected by the rubber content, the rubber crosslinking network, and the preparation process [23,24,25]. For example, Cui et al. [26] reported that in HNBR/TTPE TPVs, the melt flow rate of the TPV decreases with increasing R/P ratios. This is because as the rubber content increases, the size of the rubber particles becomes larger, and the rubber forms a stronger crosslinked network, which results in poorer processing properties of the TPVs. Kang et al. [27] reported the production of novel bio-based thermoplastic vulcanizates (TPVs) using synthesized bio-based polyester elastomers (BPEs) and poly (lactic acid) (PLA) as two components using an in situ dynamic crosslinking hybridization method. As the rubber phase increases, the mechanical properties and the hardness of the TPV decrease slightly, but its elasticity increases. As demonstrated by Wang’s group [28], differences in the ratio of the rubber phase to the plastic phase affect the morphology and mechanical properties of TPVs, and when the weight ratio of EVA/EVM TPVs is 40/60, the TPVs have excellent mechanical properties, while there is no significant phase separation at the fracture surface.

In this work, we prepare a new type of high abrasion resistance and anti-slip TPV based on an SEPS (styrene ethylene propylene styrene block copolymer), SEBS (styrene ethylene butylene styrene block copolymer), and SSBR according to dynamic sulfurization process using a torque rheometer. This material is expected to be a new material to replace the traditional vulcanized rubbers in the field of shoe soles and the automotive industry. In addition, SPES and SEBS have good overall mechanical properties, while SSBR has better anti-slip properties. This study is the first to select these two materials to be blended to prepare high abrasion resistance and high anti-slip TPVs in order to provide guidance for the recycling of high anti-slip and high abrasion resistance materials, as well as to use TPVs to replace traditional vulcanized rubber to realize the green chemical industry. Furthermore, five different SSBR/TPE composition ratios (0/100, 10/100, 20/100, 30/100, and 40/100) were selected to investigate the effect of rubber/plastic composition ratio (R/P ratio) on the abrasion resistance, rheological properties, anti-slip properties, and overall mechanical properties of TPVs. Moreover, the rubber phase morphology of dynamically vulcanized TPVs and the rubber particle morphology of abraded surfaces were investigated using scanning electron microscopy (SEM). Finally, the effects of different R/P mass ratios on various properties of TPVs based on SEPS/SEBS/SSBR were obtained to provide guidance for the preparation of TPVs with high wear resistance and anti-slip properties.

## 2. Experimental Materials and Methods

### 2.1. Experimental Materials and Equipment

SEBS TRF7512-2: Shandong Tairuifeng New Materials CO., Ltd. (Dezhou, China); SEPS YH-4051: China Petroleum & Chemical Corporation (Yueyang, China); SSBR NS560: ZS Elastomers Co., Ltd. (Tokyo, Japan). Other rubber additives, including silica (SiO_2_), zinc oxide (ZnO), tetramethylthiuram disulfide (TMTD), stearic acid, sulphur, and 2,2′-dibenzothiazoledisulfde (DM), were used without further purification. All materials are commercially available.

### 2.2. Sample Preparation

TPVs based on SEPS/SEBS/SSBR were produced using the masterbatch method [29]. The sample preparation is shown in Figure 1.

#### 2.2.1. Preparation of SSBR Masterbatch

The formulation of rubber blends was displayed using phr, which represents parts per hundred parts rubber in weight: rubber (oil-filling content 25) 125, ZnO 3.0, stearic acid 2.0, SiO_2_ 15, sulfur 0.8, DM 1.2, and TMTD 0.4.

Mixing process: The rubber blends were then mixed by an open two-roll miller (Shanghai Kechuang Rubber Machinery Equipment Co., Ltd., Shanghai, China) at room temperature (23 °C). After mixing evenly, we added zinc oxide, stearic acid, silica, DM, TMTD, and sulfur in sequence. After mixing thoroughly again, we rolled the sheet into about 2 mm test piece for spare.

#### 2.2.2. Preparation of Mixed SEPS/SEBS/SSBR

Firstly, SEBSs were added in the HAAKE torque rheometer (RM-200C, HAPU ELECTRICAL TECHNOLOGY LIMITED LIABILITY COMPANY, Harbin, China) at 70 rpm and 180 °C for 3 min; then, SEPSs were mixed for 3 min to obtain TPEs with a mass ratio of 70/30 for SEPSs and SEBSs. Thereafter, SSBR masterbatch was added to the melt and DV for 3 min. Finally, SSBR/TPE-TPVs were prepared with five different SSBR and TPE composition ratios (0/100, 10/100, 20/100, 30/100, 40/100). Immediately after the TPVs were excluded from the HAAKE torque rheometer, the TPVs were rolled through the two-roller mill at room temperature (23 °C), hot molded at 180 °C with 10 MPa, and finally cold molded at room temperature (23 °C) with a pressure of 10 MPa.

Through preliminary experiments, the effect of different mass ratios of SEPS and SEBS on TPE blends was investigated. The mechanical properties of different ratios of SEPS/SEBS are shown in Table 1.

It can be seen from the Table 1 that the comprehensive mechanical properties of the materials are relatively better when SEPS/SEBS = 70/30. By observing the tear strength and elongation at break, it was found that the introduction of the appropriate amount of SEBS can enhance the molecular flexibility of the blended materials, improve their comprehensive performance, and reduce a certain degree of hardness, which is more in line with the practical application of the subsequent materials. Meanwhile, the price of SEBS is cheaper than that of SEPS. Hence, in this paper, the SEPS/SEBS blending ratio of 70/30 was chosen for the study.

### 2.3. Characterization

#### 2.3.1. Mechanical Properties

Tensile and tear tests of materials were carried out at 23 °C using a tension machine (AI-7000M, GOTECH Testing Machine Co., Dongguan, China) at a tensile speed of 500 mm/min according to ISO 37: 2024 [30]. The hardness of the TPVs was tested at room temperature using a Shore A hardometer (LX-A, Shanghai Liuling Instrument Factory., Shanghai, China) according to ISO 48-4:2018 [31]. The resilience of thermoplastic vulcanizates was tested on a rubber resilience tester (GT-7042-RDH, GOTECH Testing Machine Co., Dongguan, China) at 23 °C according to ISO 4662:2017 [32]. All results aforementioned were the mid value of at least five measurements.

#### 2.3.2. Torque–Time Curves of DV Process

The TPVs were processed using a HAAKE torque rheometer (RM-200C, HAPU ELECTRICAL TECHNOLOGY LIMITED LIABILITY COMPANY), and the torque values of the material over time were recorded using computer software; finally, the torque–time curves were plotted using Origin (Origin 2018 64Bit) graphing software.

#### 2.3.3. Melt Flow Rate

According to ISO 1133-1:2022 [33], the melt flow rate (MFR) of TPVs was measured using melt indexer (CREE-1006, Kerui Testing Machine Co., Ltd. Shenzhen, China) at 260 °C with a 5 kg load. All results aforementioned were the average value of three measurements.

#### 2.3.4. Abrasion Resistance

The abrasion resistance of TPVs was measured using a DIN wear-resistant testing machine (GT-7012-DH1, GOTECH Testing Machine Co., Dongguan, China) according to ISO 4649:2017 [34]. All results aforementioned were the average value of three measurements.

#### 2.3.5. Morphological Analysis

The micromorphology of SEPS/SEBS/SSBR-TPV was observed using scanning electron microscope (Regulus 8100, HITACHI, Hitachi, Japan). The sample was soaked in toluene for 48 h, and the SEPSs and SEBSs were dissolved. We used a capillary tube to take a sample drop on a silicon wafer. Then, an infrared baking lamp was used to dry the sample. All specimens were sprayed with gold in order to visualize the microscopic morphology.

#### 2.3.6. Aging Resistance

The aging resistance test of thermoplastic vulcanized elastomers was carried out in Thermal Oxygen Aging Chamber (GT-7017-ELU, GOTECH Testing Machine Co., Dongguan, China) at 23 °C for 24 h. The experiments were carried out according to ISO 188:2023 [35]. All the above results are averages of three specimen measurements.

#### 2.3.7. Dynamic Mechanical Analysis (DMA)

Dynamic Mechanical Thermal Analysis of the sample was tested with a DMA tester (EPlexor 500N, NETSCH GABO Inc., Ahlden, Germany). In the tension mode, the temperature was raised from −80 °C to 80 °C at a heating rate of 3 °C/min at a frequency of 10 Hz.

## 3. Results and Discussion

### 3.1. Torque Profiles during Preparation of SSBR/TPE TPVs

Figure 1 shows the torque variation of the SEPS/SEBS/SSBR-TPVs with different R/P ratios during dynamic vulcanization. The first 480 s show that the torque reached its peak and then decreased steadily after adding SEBS and SEPS respectively, indicating that the SEBS and SEPS had completely melted and mixed evenly. The SSBR blend was added to the torque rheometer and subjected to dynamic vulcanization at high temperature for 480 s. After adding the mixed rubber, the torque decreased. This is because the rubber was heated and transformed into a molten state. After 550 s, the torque increased. This is because, at high temperatures, SSBR began to undergo a crosslinking reaction under the action of sulfur, forming a crosslinking network and leading to an increase in torque. In the final stage of compounding, the torque of the TPV decreased, as well as the crosslinked network of the rubber shears, indicating that the dynamic vulcanization of the rubber phase was complete. It could be inferred that phase reversal had occurred, and the crosslinked SSBR was broken into rubber particles and dispersed in the TPE phase under the action of shear force. By observing the curves for the R/P ratios of 0/100 and 10/100, it is clear that when the SSBR addition is 0, the torque of the material tends to be smooth after homogeneous blending. When the ratio of R/P is 10/100, the torque after adding SSBR will first surge and then decrease, then gradually rise, and reach the peak value at 600 s. This is due to the dynamic vulcanization of the rubber after the introduction of rubber into the system: the rubber produces a crosslinked network, which leads to an increase in torque.

As the amounts of SSBR rubber mix increased, the torque peak of TPVs gradually increased, and the final torque was also larger. The equilibrium torque of the thermoplastic elastomers increased gradually as the R/P mass ratio increased, demonstrating that the viscosity of the thermoplastic elastomers in the molten state also increases gradually, and the crosslinked rubbers are more viscous and have more influence on the increase in the melt viscosity of thermoplastic elastomers. It has been claimed that when the R/P ratio is higher, then the higher the viscosity of the thermoplastic elastomer, the more difficult it is for the rubber to be broken during the dynamic vulcanization process, which ultimately results in larger rubber particle sizes [36].

### 3.2. Mechanical Properties of SSBR/TPE TPVs

Figure 2 shows the mechanical properties of the SSBR/TPE TPVs with different SSBR contents. As the amount of rubber added increased, the tensile strength and tear strength of the TPVs decreased. It can be inferred that rubber and TPE form an island structure in the TPV, and the rubber particles form fracture points in the continuous phase of the TPE. As the rubber content increases, the volume of the island phase continues to increase, the degree of rubber dispersion decreases, and there is a reduced amount of continuous phases in thermoplastic elastomers, resulting in a decrease in the tensile strength and tear strength of the TPVs. With the increase in the rubber amount, the rubber particles in the TPV undego agglomeration and other phenomena, and the increase in the agglomeration of these rubber particles will yield the physical properties of the TPV such as tensile strength, tear strength, and other physical properties, which showed a decreasing trend. Due to the poor hardness and rebound performance of the SSBR rubber blend, as the rubber content increased, the hardness and rebound rate of the TPVs both decreased to a certain extent. Typically, the elasticity of a thermoplastic vulcanizate elastomer is provided primarily by crosslinked rubber, which places higher demands on the content and crosslinking degree of the rubber phase. However, the plastic phase consisting of SEPS/SEBS in this paper had better elasticity, so the elasticity was provided by the plastic phase in the SEPS/SEBS/SSBR TPVs. At the same time, with the increase in rubber particles, the intermolecular interaction force of the TPV decreases, and the force used to overcome the viscous resistance between molecules on the rubber also decreases, resulting in a decrease in the material’s rebound rate.

### 3.3. Compression Permanent Deformation of SSBR/TPE TPVs

The TPE in this paper was composed of SEPSs and SEBSs. They are styrenic thermoplastic elastomers. SEBS is a linear triblock copolymer with polystyrene as the terminal segment and ethylene-butene copolymer obtained by the hydrogenation of polybutadiene as the middle elastic block. SEPS is a linear triblock copolymer with polystyrene as the terminal segment and ethylene-propylene copolymer obtained by the hydrogenation of polyisoprene as the middle elastic block. The compatibility of the two materials is better, and the material’s tensile strength, tear strength, resilience, and hardness are relatively higher, but because of the two-dimensional linear structure, when the external force and time exceeds the limit, the linear molecular chain appears to be relatively slippery, the material occurs in compression permanent deformation. But after dynamic vulcanization, the rubber forms a crosslinked network. When the rubber content increases, the three-dimensional crosslinked network makes the molecular chain relatively less easy to slip, and the compression permanent deformation becomes smaller. By observing Figure 3, it can be seen that as the rubber content in SSBR/TPE TPVs increased, a stronger crosslinked network was formed in the TPVs, which makes them less prone to deformation after external compression. As the rubber content increased, the compression permanent deformation became significantly smaller. Observing the compression permanent deformation for SSBR/TPE mass ratios of 0/100 and 10/100, it can be seen that in the addition of SSBR to the TPE, the compression permanent deformation of the TPVs decreased to about 15%, indicating that the SSBR/TPE TPVs prepared in this work are high elastic elastomers based on ASTM D1566-15 [37].

### 3.4. Rheological Properties of SSBR/TPE TPVs

Unlike traditional thermosetting vulcanized rubber, TPV has good processability and recyclability, and its rheological properties are of great guiding significance for the preparation of TPV materials [38]. If TPVs have good rheological properties in a molten state, then the material has good processing performance, which is conducive to recycling and improving its preparation efficiency [13,39]. The influence of different R/P ratios on the rheological properties of SSBR/TPE TPVs was systematically studied using the melt flow rate (MRF) test. As shown in Figure 4, with the increase in the R/P ratio, the rheological properties of the SSBR/TPE TPVs deteriorated. The higher the rubber content, the worse the processing flowability of TPVs in the molten state, and the greater the difficulty of the forming process. The increase in the rubber phase density and size leads to the formation of a stronger rubber crosslinked network in the thermoplastic elastomer, which makes the rheological properties of the thermoplastic elastomer worse.

### 3.5. Wear Resistance of SSBR/TPE TPVs

The DIN wear volumes of SSBR/TPE TPVs with different R/P ratios are shown in Figure 5, indicating that as the R/P ratio increases, the wear volume gradually increases. This is because the increase in SSBR content leads to a continuous increase in the volume of the rubber particles in the system, resulting in the formation of small particles on the friction surface of the rubber. These particles experience friction and fall off during the friction process, thereby increasing the wear of the material. Furthermore, an increase in the rubber content may also lead to agglomeration, resulting in a decrease in the intermolecular forces and a decrease in the flexibility of the molecular chain segments, which may also contribute to the ultimate decrease in material wear performance.

Observing the surface morphology after abrasion through Figure 6, it can be found that the abrasion surface appears to be hollow after the rubber particles fell out, and the number and size of these hollows increased with the increase in the R/P mass ratio. This is because both the size and the density of the rubber particles increase with the amount of rubber, and the size of the hollows left on the abrasion surface increases with the size of the hollows after the fall out of the rubber particles occurs.

### 3.6. Morphology of SSBR/TPE TPVs

Figure 7 shows the morphology of the rubbery phase of SSBR/TPE with different R/P ratios observed using a scanning electron microscope. Before SEM observation, we soaked an appropriate amount of SSBR/TPE TPVs in toluene to dissolve the TPE and observe the crosslinked SSBR particles. From Figure 7, it can be seen that as the R/P ratio increased, both the size and density of the rubber particles increased [40]. At the same time, in the case of a large amount of rubber addition, the SSBR particles also undergo aggregation, leading to a further increase in their size. The increase in the rubber phase size leads to an increase in the strength of its rubber crosslinking network, a decrease in deformation ability, and a deterioration in the TPV rheological properties, which is consistent with the conclusion in Section 3.4. At the same time, the rubber particles become larger, making them more prone to wear during DIN wear, which is consistent with the conclusion in Section 3.5.

### 3.7. Dynamic Mechanical Analysis of SSBR/TPE TPVs

Figure 8 shows the DMA of TPVs with different R/P mass ratios. It can be seen that after adding SSBR, there are two obvious peaks in the DMA image, with the peak around 0 °C displayed by the SSBR. The tan δ at 0 °C can reflect the anti-slip performance of the material [41]. It is evident from Figure 7 that the addition of SSBR significantly increased the tan δ of the material at 0 °C, and as the amount of SSBR added increased, the tan δ value at 0 °C also increased. This indirectly reflects that the larger the R/P ratio, the better the slip resistance of SSBR/TPE TPVs.

### 3.8. Skid Resistance of SSBR/TPE TPVs

According to the friction theory model, the adhesion friction and hysteresis friction are related to the tan δ value, and the more energy that is lost in the friction process, the larger the value of tan δ and the larger the friction coefficient [42,43,44]. Figure 9 shows the static friction force, the dynamic and static friction, and the coefficients of the dynamic and static friction of the SSBR/TPE TPVs at different R/P ratios. It can be clearly seen that with the increase in rubber content, the static friction force, dynamic friction force, static friction coefficient, and dynamic friction coefficient values of the TPVs were gradually improving. Adding SSBR to TPE can greatly improve its anti-slip performance, which is consistent with the conclusion of Section 3.7. TPE itself does not have good slip resistance. By adding an SSBR with excellent slip resistance, the blend material can be transformed into a thermoplastic elastomer material with excellent slip resistance.

### 3.9. Secondary Processing Performance of SSBR/TPE TPVs

The comprehensive performance retention rate after secondary processing is an important indicator for evaluating TPVs, and a high-performance retention rate can only mean that the material can be recycled [1,45,46]. We added the TPVs after use to an HAAKE torque rheometer (RM-200C, HAPU ELECTRICAL TECHNOLOGY LIMITED LIABILITY COMPANY) with 70 rpm at 180 °C for 4 min. Immediately after the co-mixture was excluded from the HAAKE torque rheometer, the TPVs were rolled through a two-roller mill at room temperature, then hot molded at 180 °C with a pressure of 10 MPa, and finally cold molded at room temperature with a pressure of 10 MPa. Figure 10 shows that there is no obvious difference in the appearance of the samples before and after secondary processing. Through the DMA test before and after the secondary processing shown in Figure 11, it can be seen that before and after the secondary processing, the dynamic mechanical analysis revealed almost no change. The performance test results are shown in Table 2. The tensile strength, tear strength, and hardness retention of the TPVs after secondary processing were all above 93%, indicating that the SSBR/TPE TPVs prepared using dynamic vulcanization conform to the repeatable processing characteristics of thermoplastic elastomers. The blended material prepared by this process can be recycled and reused.

## 4. Conclusions

In summary, composite TPVs of SSBR, SEPSs, and SEBSs were successfully prepared through dynamic vulcanization using a torque rheometer, and the influence of the R/P mass ratio on the various properties and morphologies of SSBR/TPE composite TPVs was systematically studied. As the R/P ratio increased, the tensile strength and the tear strength of TPVs slightly decreased, but their resistance to compression permanent deformation increased. Combined with the morphology study (SEM), it was found that the size of the rubber particles increased significantly with higher R/P mass ratios, which enhanced the rubber crosslinking network but reduced the rheological properties of the TPVs. The increase in rubber particle size and decrease in dispersion uniformity also led to a higher DIN wear volume. Additionally, DMA and anti-slip performance tests demonstrated improved anti-slip properties with increasing R/P mass ratios: when the R/P ratio was 40/100, the tan δ of the TPVs at 0 °C was 0.35. The comprehensive mechanical property retention rate of the TPVs after secondary processing reached more than 93.9%. This study contributes to understanding the relationship between the component ratio of rubber–plastic blends and the properties of thermoplastic vulcanized elastomers. It provides theoretical and industrial guidance for developing TPE-based high wear-resistant and high slip-resistant thermoplastic elastomers and promotes the use of TPVs in high wear-resistant and high slip-resistant fields, such as shoe materials.

## Data Availability

Our team’s data are accurate and reliable, and the experiments are repeatable. We guarantee that the information is fully available. Data are contained within the article.

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
