# Peer review of "Thermoplastic Vulcanizates with an Integration of High Wear-Resistant and Anti-Slip Properties Based on Styrene Ethylene Propylene Styrene Block Copolymer/Styrene Ethylene Butylene Styrene Block Copolymer/Solution-Polymerization Styrene-Butadiene Rubber"

_polymers, 2024, doi:10.3390/polym16152221_

Round 1

Reviewer 1 Report

Comments and Suggestions for Authors

1.             3-min mixing for blending rubber masterbatch with TPE is not enough to complete DV. 

2.             Please provide the full name of SEBS and SEPS.

3.             Why the authors use SSBR? How about ESBR?

4.             Please provide Mooney viscosity, MW, MWD, styrene content, block segments of all polymers used in this work.

5.             Please provide equation to calculate volume loss from DIN abration.

6.             What is contact angle of the sample? What is the relation between contact angle and abrasion resistance?

7.             Why the authors use two polymers for TPE? Why do not use only SEBS or only SEPS? The poor compatibility of 3 polymers in TPV can lead to poor physical properties.

8.             What is novelty of this work? The result in this work has no significant contribution to polymer science.

9.             What is abrasion resistance after reprocessing?

10.       Why the authors performed aging resistance at 23oC for 24 h.

11.       How to perform compression permanent deformation of TPV?

12.       Why resilience reduces with increasing SSBR?

13. What is indication that this TPV shows excellent abrasion resistance?

Author Response

Thank you very much for your kind consideration and the your comments on our manuscript.We would like to express our great appreciation on your perceptive review.We have revised the manuscript carefully according to the reviewers’ comments in full. Point-by-point responses are enclosed the attachment. Please see the attachment.Once again, thank you.

Reviewer 2 Report

Comments and Suggestions for Authors

This study utilizes dynamic vulcanization to develop new thermoplastic elastomers from solution-polymerized SSBR and SEPS/SEBS. The high slip and abrasion resistance of SSBR enhances the low slip resistance of SEPS/SEBS, providing a potential solution for recycling shoe sole materials. The paper explores the impact of varying rubber/plastic phase ratios on the properties of SSBR/TPE TPVs. The article contributes to research in the field but it lacks more specific details on innovations. Please address the following points for corrections:

1.      The Abstract discusses general trends, such as the decrease in tensile strength, tear strength, hardness, and resilience with an increasing R/P ratio. However, specific numerical values are needed to support these claims.

2.      In the Abstract section, there are grammatical and spelling errors, such as the use of "SBES" instead of "SEBS".

3.      The Introduction provides an overview of the general research area but it does not clearly state the specific objectives and novelties of the current study.

4.      Use the following papers. Advancing sustainable shape memory polymers through 4D printing of polylactic acid-polybutylene adipate terephthalate blends. Design, processing, 3D/4D printing, and characterization of the novel PETG–PBAT blends.

5.      The sentence in the Introduction section, "The rubber particles are dispersed in the continuous resin matrix under the strong shear, forming a "sea-island" phase structure" could be rephrased for clarity and conciseness.

6.      The Introduction section mentions "styrene-based thermoplastic elastomers (SBS, SIS, SEBS, SEPS)" but then refers to "SEPS, SEBS and SSBR" without explaining the relationship between these materials.

7.      In the Preparation of mixed SEPS/SEBS/SSBR section, the use of terms like "co-mixture" and "thermo-molded" may not be familiar to all readers and could be better explained.

8.      In the Characterization section, it is essential to provide detailed explanations about test accuracy, errors, numbers, and verification methods.

9.      In Section 3.2 it is mentioned that "the elasticity of a thermoplastic vulcanizate elastomer is provided primarily by crosslinked rubber, which places higher demands on the content and crosslinking degree of the rubber phase." However, it then states that "the plastic phase consisting of SEPS/SEBS in this paper has better elasticity, so the elasticity is provided by the plastic phase in SEPS/SEBS/SSBR TPVs." This statement appears contradictory and needs additional clarification or explanation.

10.  Section 3.6 of the article incorrectly refers to "Figure 8" when it should actually be referring to "Figure 7".

11.  The Conclusions section could be enhanced by including quantitative data to better reflect the research achievements.

Comments on the Quality of English Language

***

Author Response

Thank you very much for your kind consideration and the your comments on our manuscript.We would like to express our great appreciation on your perceptive review.We have revised the manuscript carefully according to the your comments in full. Point-by-point responses are enclosed the attachment. Please see the attachment.Once again, thank you.

Reviewer 3 Report

Comments and Suggestions for Authors

1. Did the authors only use SEPS/SEBS = 70/30. What is the reason?

2. There was no the unit of ingredients such as “ZnO 3.0, stearic acid 2.0, SiO2 15, sulfur 0.8, DM 1.2, TMTD 0.4”.

3. What is the unit of “r/min”?

4. There were a lot of typos and grammatical errors in this manuscript.

5. The abbreviations of second and minute are ‘s’ and ‘min’, respectively.

6. Figure 1: There was no detailed discussion about the SSBR/TPE = 0/100 and 10/100.

7. Figures 2 and 3: The physical properties in Figures 2 and 3 showed all the same trends. What is the reason? Although the SSBR/TPE = 0/100 sample had the highest tensile strength, tear strength, resilience, and hardness, its compression deformation was larger than those of the others. What is the reason?

8. Figure 5: By adding SSBR to TPE, the wear property became worse. This is a fatal weak point for outsole. Do the authors have any improvement proposal?

Comments on the Quality of English Language

There were a lot of typos and grammatical errors in this manuscript.

Author Response

Thank you very much for taking the time to review this manuscript.Thank you for your very
kind comments on this study! I've made some major changes to the article based on your
comments, and I'm honored to have you review it!

The point-by-point responses are attached as an annex. Please see the attachment. Thank you all again.

Round 2

Reviewer 1 Report

Comments and Suggestions for Authors

The author do not show significant difference from original paper.

Author Response

Thank you for your comments. We have made significant changes to the article based on all the reviewers' comments. Point-by-point responses are included in the attachment. I have marked all changes in red, blue or green in the original article. Thanks again to everyone!

Reviewer 2 Report

Comments and Suggestions for Authors

Accept in present form.

Author Response

感谢您的指导。

Reviewer 3 Report

Comments and Suggestions for Authors

1. About the response 1: The authors gave the mechanical properties of the samples SEPS/SEBS = 0/100, 30/70, 50/50, 70/30, and 100/0, and discussed the results as the reply. However, these contents were not included in the manuscript. For the sake of readers’ understanding, the authors should include the physical properties of the samples SEPS/SEBS = 0/100, 30/70, 50/50, 70/30, and 100/0.

2. About the response 5: There were still ‘second’ and ‘minute’ in the manuscript. The authors should use ‘s’ and ‘min’ as the units of ‘second’ and ‘minute’.

3. About the response 7: The authors discussed the results in Figures 2 and 3 as the reply. However, these contents were not properly reflected in the manuscript. The authors should properly reflect these contents in the manuscript.

Comments on the Quality of English Language

...

Author Response

Thank you for your guidance. I apologize for not understanding your first comment accurately. I have revised the article according to your guidance. See below for an  Point-by-point response.

Comments 1: About the response 1: The authors gave the mechanical properties of the samples SEPS/SEBS = 0/100, 30/70, 50/50, 70/30, and 100/0, and discussed the results as the reply. However, these contents were not included in the manuscript. For the sake of readers’ understanding, the authors should include the physical properties of the samples SEPS/SEBS = 0/100, 30/70, 50/50, 70/30, and 100/0.

Response 1: Thank you for your guidance. As per your suggestion, I have added the relevant content in Section 2.2.2. The revised portions are marked in green in the revised manuscript. Once again, I express my gratitude to you.

Comments 2: About the response 5: There were still ‘second’ and ‘minute’ in the manuscript. The authors should use ‘s’ and ‘min’ as the units of ‘second’ and ‘minute’.

Response 2: Thank you for your guidance. I checked the article based on your suggestions, and use ‘s’ and ‘min’ as the units of ‘second’ and ‘minute’. The revised portions are marked in green in the revised manuscript. Once again, I express my gratitude to you.

Comments 3: About the response 7: The authors discussed the results in Figures 2 and 3 as the reply. However, these contents were not properly reflected in the manuscript. The authors should properly reflect these contents in the manuscript.

Response 3: Thank you for your guidance.I've revised Sections 3.2 and 3.3 based on your suggestions. The revised portions are marked in green in the revised manuscript. Once again, I express my gratitude to you.

Round 3

Reviewer 1 Report

Comments and Suggestions for Authors

I can be published after revision

Reviewer 3 Report

Comments and Suggestions for Authors

The authors properly responded to my comments.